# Utilization and Effect of Apple Pomace Powder on Quality Characteristics of Turkey Sausages

**DOI:** 10.3390/foods13172807

**Published:** 2024-09-04

**Authors:** Aigerim Koishybayeva, Malgorzata Korzeniowska

**Affiliations:** 1Department of Food Technology, Almaty Technological University, Almaty 050000, Kazakhstan; aigerim.koishybayeva@atu.edu.kz; 2Department of Functional Food Products Development, Wroclaw University of Environmental and Life Sciences, 50-375 Wroclaw, Poland

**Keywords:** physico-chemical properties, texture profile analysis, apple pomace, cooked sausages, functional food

## Abstract

The present study was conducted to develop turkey sausages by incorporating freeze-dried apple pomace (FDAP) at 3, 5, and 8% by replacing turkey breast meat. Three sausage formulations and the control of turkey sausages were subjected to physicochemical properties: proximate content, water-holding capacity (WHC), cooking yield, pH, color, textural parameters, antioxidant activity, and microbiological and sensory properties. The parameters were analyzed during storage from days 0 to 7. The addition of FDAP to turkey sausages resulted in a significant (*p* ≤ 0.05) decrease in moisture and protein contents, whereas no significant difference was found in fat and ash contents. The increased incorporation of FDAP resulted in decreased pH, cooking loss, lightness, redness, and yellowness in turkey sausages, whereas an increase in total phenolic content, fiber content, ABTS, and DPPH values was observed. FDAP, as a low-cost source of valuable phenolic content, strongly inhibited microorganism growth during the storage of turkey sausages. The sensory scores of turkey sausages containing 3% FDAP for other sensory traits, such as flavor, texture, juiciness, tenderness, and overall acceptability, were comparable to those of the control. Scores for sensory attributes declined significantly with a further increase in FDAP in turkey sausages. It is concluded that turkey sausages with very good acceptability can be prepared by incorporating freeze-dried apple pomace each at a 3% level.

## 1. Introduction

Apples, particularly the domestic variety (*Malus domestica* L.), are the world’s most widely consumed fruit, with a global production of 95.8 million tons in 2022, including 267 thousand tons produced in Kazakhstan [1]. During the industrial extraction of apple juice, a significant portion of the raw material is wasted in the form of apple pomace, which constitutes 25–30% of the total processed fruit [2].

Apple pomace has been identified as a significant source of important nutrients and phytochemicals, including carbohydrates, vitamins, and minerals [3]. According to a study by Bhushan et al. [4], apple pomace containing residual fruit parts has a higher fiber content, ranging from 4.4 to 47.3 g of fiber per 100 g. It should be noted that apple pomace has a substantial amount of polyphenols, ranging from 31% to 51% [3]. Apple pomace contains natural antioxidants with strong antioxidant activity, such as quercetin glycosides, phloridzin, and other phenolic constituents [5,6].

Several studies have revealed the health benefits of consuming apple pomace in both humans and animals. For example, apple pomace fiber, or pectin, has been shown to improve intestinal health [7]. Additionally, it enhances the sensitivity to insulin, cholesterol levels, and functionality of the gut microbiota [8], and contributes to antioxidant defense systems [9]. Apple pomace has also been linked to reductions in body weight and improvements in serum lipid profiles [10], as well as enhanced gastrointestinal health due to the presence of flavonoids, which improve local interactions within the digestive tract [11,12]. Furthermore, apple pomace consumption has been associated with favorable reductions in gut enzymes without affecting nitrogen utilization [13] and improvements in blood glucose levels [11,14].

For example, studies have shown that apple pomace can increase the crude fiber content in chicken patties, chicken nuggets, and low-fat chicken nuggets [15,16,17]. Furthermore, apple pomace has been found to improve the emulsion stability of mutton nuggets, buffalo meat emulsion, and patties [18,19,20], and to increase the antioxidant activity of Italian salami and baked meat products [21,22]. However, there is currently no scientific information available on the use of apple pomace in turkey sausages. This study aimed to investigate the potential of incorporating apple pomace into turkey sausage production, providing the dual benefits of improving the nutritional profile of sausages and reducing food waste. Turkey sausages were produced by adding freeze-dried apple pomace (FDAP) to the formulation at levels of 0%, 3%, 5%, and 8% in place of turkey breast meat. These sausages were then evaluated for cooking yield, water-holding capacity (WHC), pH, proximate composition, texture, color, microbiological analyses, and sensory properties.

## 2. Materials and Methods

### 2.1. Apple Pomace Powder Preparation

Apples were procured from a local establishment (Zeleniy bazar, Almaty, Kazakhstan) and washed with warm, running water. The cores of the apples were then excised, and the pomace was extracted by juicing the fruit. The apple pomace was subsequently freeze-dried using a Free Zone12 freeze dryer (Labconco Corporation, Kansas City, MO, USA), a process that lasted for three days after being frozen at a temperature of −80 °C. During the drying process, a pressure of 0.04 mbar was maintained. The freeze-dried apple pomace was then ground in a colloid mill to a powder, with the resulting product having the composition and physical properties summarized in Table 1. The powder was then placed in polyethylene bags and stored at 4 °C for future use.

### 2.2. Sausage Preparation

Turkey meat and skin were purchased from a local processor (Gizewski Ltd., Slupia Kapitulna, Poland). The approximate chemical composition of turkey meat and skin was 74.33% moisture, 24.52% protein, 0.91% fat, and 1.08% ash, and 52.7% moisture, 10.28% protein, 24.63% fat, and 0.47% ash, respectively. Non-meat ingredients included curing salt, isolated soy protein, sugar, and FDAP. Before preparing the turkey sausages, all visible skin, connective tissue, and fat were trimmed from the meat. Deboned and frozen turkey meat and skin were thawed in a refrigerator (4 ± 1 °C) and ground in an electrical meat mincer (W-82AN Spomasz, Zary, Poland) using 8 mm and 4 mm plates, respectively. The following formulations were obtained: C—control sample (0% FDAP); T1—sausages with a 3% addition of FDAP; T2—sausages with a 5% addition of FDAP; and T3—sausages with an 8% addition of FDAP (Table 2). After adding the non-meat ingredients to the minced meat, the mass was homogenized (9000 rpm) using a Büchi Mixer B-400 (BÜCHI Labortechnik GmbH, Essen, Germany) for 30 s to prepare a stable emulsion (the final temperature of emulsion varied around 10 °C). Then, the raw emulsion was stuffed into 60 g polypropylene tubes with a capacity of 50 mL (2.5 cm diameter and 12 cm height). The sausages were then heat-processed in a temperature-controlled water bath (Julabo TW12, Julabo Inc., Allentown, PA, USA) maintained at 80 °C using a thermocouple inserted into the center of a link to an internal temperature of 72 °C. After cooking, the sausages were cooled on ice to room temperature and stored at refrigeration temperatures for further use.

### 2.3. pH

The turkey sausages were stored at 4 °C, and pH was measured on days 0, 3, 5, and 7 in a homogenate prepared with 5 g of sample and distilled water (20 mL) using a pH meter electrode (Orion 3-Star pH Benchtop Meter, Thermo Fisher Scientific Inc.) at room temperature. All measurements were performed in triplicate.

### 2.4. Cooking Yield Measurement

Cooking yield was assessed by measuring the weight of turkey sausage samples both before and after cooking for each treatment using an electronic weighing balance. The ratio of cooked weight to raw weight was then calculated and expressed as a percentage.

### 2.5. Water Holding Capacity Analysis

The water-holding capacity of turkey sausages was determined using a previously described method [23]. Ten grams of the sample was placed in a tube, mixed with 40 mL of distilled water, and kept in a water bath at 30 °C for 30 min, followed by centrifugation at 3000 rpm for 30 min. The results were calculated as percentages using the following Equation (1):
(1)
$$WHC=\frac{Weight\;of\;sample\;after\;removing\;supernatant}{Weight\;of\;sample\;mixed\;with\;distilled\;water}*100$$


### 2.6. Proximate Composition

Homogenized turkey sausage samples (three containing FDAP and the control) were analyzed for moisture content by weighing the loss after 12 h of drying at 105 °C in a convectional dryer, the fat content by the Soxhlet method with a solvent extraction system (Universal Extractor E-800, BUCHI, Essen, Germany), the protein content by the Kjeldahl method with a nitrogen analyzer, a Kjeltec TM 2300 (FOSS, Hilleroed, Denmark), and the ash according to the AOAC Official Methods of Analysis (17th edition) by the Association of Official Analytical Chemists (muffle furnace) [24]. The total dietary fiber (TDF) was determined using the standard AOAC Official Methods of Analysis, 16th Edition [25].

### 2.7. Color Measurements

The color parameters, which were expressed as the L (lightness), a (redness), and b (yellowness) values of the turkey sausage samples, were determined using a reflectance colorimeter (Minolta CR-400 calibrated with a white plate, L = +97.83, a = −0.43, b = +1.98). Turkey sausage samples were cut into slices (15 × 25 mm) before each measurement. The analysis was carried out directly after production on day 0 and repeated after day 3, day 5, and day 7 of chill storage. Analyses were performed in triplicate in each series (3 × 3).

### 2.8. Texture Profile Analysis

Texture Profile Analysis (TPA) of turkey sausages was performed according to the procedure outlined by Bourne [26] using a Zwick/Roell Z010 testing machine (Zwick Testing Machines Ltd., Leominster Herefordshire, UK). Three slices of each sample (length and diameter of 15 and 25 mm, respectively) were compressed twice to 50% of their original height. The conditions of the texture analysis were as follows: 50% deformation, head speed of 60 mm/min, and relaxation time of 30 s. The parameters determined were hardness (kg); adhesiveness (kg × s); cohesiveness (mm/mm); springiness (mm); gumminess (mm): (hardness × cohesiveness); and chewiness (kg): (springiness × gumminess). TPA was performed on three replicates of each sample at room temperature (22 ± 1 °C) directly after the production process and at 0, 3, 5, and 7 days of chill storage.

### 2.9. Antioxidant Activity

The determination of 2,2′-azinobis (3-ethylbenzothiaziline-6-sulfonate) (ABTS+) radical scavenging activity of turkey sausages was slightly modified from that reported by Bai et al. [27]. ABTS+ was produced by mixing 7 mM stock solution with 2.45 mM K_2_S_2_O_8_, followed by incubation in the dark at 25 °C for 12–16 h. Then, 990 μL of ABTS+ solution was mixed with 10 μL of sausage supernatant and incubated at an ambient temperature (~25 °C) for 6 min. The control comprised 990 μL ABTS + solution mixed with 70% EtOH (10 μL). The absorbance was measured spectrophotometrically at 734 nm.

The determination of the 1,1-diphenyl-2-pierylhydrazy (DPPH) radical scavenging activity of the turkey sausages was slightly modified from that of Yen and Chen [28]. Aliquots (20 μL) from the sausage supernatant were vigorously mixed with 200 μL 0.3 mM of ethanolic DPPH radical solution by vortex for 1 min and subsequently kept in the dark for 30 min at an ambient temperature (25 °C). The absorbance was recorded against a blank at 517 nm using a UV-Vis Spectrophotometer (GENESYS™ 180, Thermo Fisher Scientific Inc., Waltham, MA, USA). DPPH radical scavenging activity values were reported as mM Trolox.

### 2.10. Total Phenolic Evaluation

Total polyphenol content was determined according to the method of the LC-MS [29]. FDAP (1 g) was weighed and placed in a test tube; 20 mL of 80% aqueous methanol with 1% hydrochloric acid was added and mixed. The test tubes were then sonicated twice for 15 min and left at room temperature in the dark for 24 h (20 °C). The resulting extract was centrifuged at 20,000× *g* during 10 min and the supernatant was collected at 4 °C. Polyphenolic compounds were quantified using a Merck-Hitachi HPLC system (Tokyo, Japan) with a diode array detector. Separation was performed on a Synergi Fusion RP-80A column (Phenomenex, Torrance, CA, USA) at 30 °C, using 2.5% acetic acid (solvent A) and acetonitrile (solvent B). A gradient from 0% to 25% B over 36 min was used, with detection at 280, 320, and 360 nm for various polyphenol classes. The amount of the total phenolic compounds was expressed as gallic acid equivalent (GAE) [mg GAE/kg] based on the standard curve.

### 2.11. Microbiological Analysis

The turkey sausages were analyzed after their production (day 0) and this was repeated after day 7 by the methodology used for the microbiological analysis, as described by the authors in a previous study [30]. The total number of microorganisms was evaluated using the pour-plate method. Therefore, 20 g of sample was put into sterile polyethylene bags and homogenized with peptone water (peptone 1.0 g/L, NaCl 8.5 g/L) for 1 min using a laboratory blender (Easy Mix, AES Laboratories, Paris, France). Afterward, a series of dilutions were prepared, and 1 mL of each dilution was placed in the center of Petri dishes (10 cm diameter). Subsequently, 15 mL of culture medium that consisted of 5.0 g/L tryptone, 2.5 g/L yeast extract, 18.0 g/L agar (BTL Ltd. Lodz, Poland), and 1.0 g/L anhydrous glucose (Chempur, Piekary Slaskie, Poland) was added to each plate and mixed carefully [31,32]. The plates were then incubated at 30 ± 1 °C for 72 h. Colonies of microorganisms were counted using a hand-type colony counter (N.USUI & Co., Ltd., BIO KOBE, Kobe, Japan) and expressed as the total number of microorganisms in 1 g of the sample.

### 2.12. Sensory Evaluation

The sensory attributes of turkey sausages were evaluated using a scale from 1-dislike extremely to 9-like extremely [33]. Turkey sausage samples were cut into 1 cm slices and served warm to eight previously trained panelists in the Department of Functional Food Products Development (Wroclaw University of Environmental and Life Sciences, Poland), whose ages ranged from 24 to 35 years. Each sample of turkey sausage was coded with a randomly selected 3-digit number and served on a white paper plate. Coded samples of turkey sausages were presented to test panelists individually, and they were asked to evaluate the samples for sensory attributes such as flavor, color, taste, texture, juiciness, and overall acceptability. Room temperature water was provided to clean the palate between samples, and sensory evaluation was performed before a midday meal.

### 2.13. Statistical Analysis

All assays were performed in at least three replicates and three measurements (3 × 3). The results are presented as mean and standard error (mean ± standard error). The differences between the mean values were considered significant at *p* ≤ 0.05 by using Duncan’s multiple range tests. The data were analyzed using one-way (between groups at the same time intervals) and two-way (within a group between individual time intervals) analyses of variance (ANOVA). All analyses were performed using Statistica software ver. 13.0 (StatSoft Inc., Cracow, Poland).

## 3. Results and Discussion

### 3.1. Proximate Composition

The proximate composition of turkey sausages formulated with different levels of FDAP is shown in Table 3. The moisture and protein contents of the turkey sausages with increasing levels of FDAP were lower than those of the control (*p* ≤ 0.05). Among all FDAP-based treatments, turkey sausages that were replaced with 8% (T3) pomace had significantly lower (*p* ≤ 0.05) moisture (68.71%) and protein (13.89%) values than those of the control, which had the highest moisture (70.93%) and protein (16.87%) values. The observed decrease in moisture content in the treated turkey sausages, particularly in treatment T3 (68.67%), can be attributed to the inclusion of freeze-dried apple pomace (FDAP) at increasing levels. FDAP had a much lower moisture content (8.91%) compared to turkey meat, which results in a reduction of overall moisture when it partially replaced the meat in the sausages. This was consistent with the results obtained by Yadav et al. [34] and Huda et al. [18], who reported that the moisture content of dietary-fiber-enriched chicken sausages and the protein content of treated mutton nuggets decreased with increasing levels of dried apple pomace. There was no statistical difference in fat content between the control and FDAP-treated turkey sausages. As the FDAP level increased, fat content also increased slightly (*p* ≥ 0.05) at each level. These results are in agreement with the data reported by Younis and Ahmad [20] with respect to the absorbed melted fat due to its good oil-holding capacity in buffalo meat patties. The ash content of the turkey sausage samples containing FDAP was significantly lower than that of the control without FDAP (*p* ≤ 0.05). The highest and lowest values of ash content were found in the control (2.32) and T1 (1.87%), respectively. These results aligned with the findings of Verma et al. [17], who observed a reduction in the ash content of low-fat chicken nuggets as the levels of apple pulp increased. The incorporation of FDAP in turkey sausages led to an increase in the dietary fiber content (Figure 1) of treated turkey sausages from 1.18 to 1.63% compared with those in the control, which contained no dietary fiber. Similarly, the fiber content in treated meat products was significantly higher than that in the control, with the fiber content increasing proportionally with the level of apple pomace used [17,18,19].

### 3.2. Water Holding Capacity

The results of the Water Holding Capacity (WHC) analysis indicated that the addition of FDAP to turkey sausage formulations did not have a significant interactive effect (*p* ≥ 0.05) on the WHC (Table 3). Among the formulations, the WHC value of T1 with 3% FDAP was the lowest (74.03%) compared to that of the control (75.93%), T2 (75.13%), and T3 (76.40%). Although there was a slight increase (*p* ≥ 0.05) in the WHC with higher levels of FDAP, the difference was not statistically significant. The stable trend observed in our study may be attributed to the specific source and chemical properties of the dietary fiber used, as suggested by Elleuch et al. [35]. The affinity of dietary fiber to bind water can vary greatly, depending on its source and chemical composition. Contrary to our findings, other studies have reported a significant increase in WHC with the addition of dietary fiber. For instance, authors [20,36,37] observed significant enhancements in the WHC of chevon rolls, Irish breakfast sausages, and buffalo meat patties.

### 3.3. Cooking Yield

The cooking yield showed a slight increase, similar to that of WHC, with the incorporation of FDAP into turkey sausages. The control sausages without FDAP exhibited no significantly higher (*p* ≥ 0.05) cooking yield values compared to the treated turkey sausages with added FDAP. The cooking yield of meat products is generally attributed to the ability of the protein matrix to retain water and bind fat [38,39]. The lower protein content among turkey sausages treated with FDAP (3–8%) might have contributed to the reduced cooking yield. Similar trends in cooking yield have been reported in other studies, such as that by Verma et al. [17], in which different levels of apple pulp were added to low-fat chicken nuggets.

### 3.4. pH Value

It is possible to observe that the pH of the turkey sausages formulated with FDAP was significantly (*p* ≤ 0.05) lower than that in the control without FDAP (Table 4). The pH value of the control turkey sausages was relatively stable and increased slightly from 5.80 (day 0) to 5.85 (day 7). Throughout the storage period of sausages containing FDAP, their pH value was significantly lower than that of the control (on average 5.63, 5.50, and 5.34 for T1, T2, and T3, respectively) and depended on the dose of the additive used (T1 < T3). A significant effect was observed in the turkey sausage containing 3% (T1) and 8% (T3) FDAP; the pH values from day 0 to day 7 varied between 5.63 to 5.59, and 5.34 to 5.31, respectively. This may be explained by the presence of organic acids in apple pomace. A similar result was reported by Verma et al. [17], who observed a decrease in the pH of chicken nuggets. Keska et al. [22] also reported a similar trend in baked meat products; the incorporation of freeze-dried apple pomace in baked meat products resulted in a gradual decrease in product pH as the level of FDAP increased. Thus, the obtained results indicated that the pH values of the treatments decreased with increasing FDAP levels.

### 3.5. Antioxidant Activity

The ABTS and DPPH values of the turkey sausages increased significantly with increasing levels of FDAP (Table 3). All values increased with respect to the control (*p* ≤ 0.05), especially when the percentage of FDAP was 8%; in fact, the TPC increased from 18.80 to 78.24 mg GAE/kg (Figure 2), ABTS from 2.49 to 4.06 mg TE/g, and DPPH from 4.0 to 7.02 mg TE/g. Treatment 3 (with 8% FDAP) resulted in a higher content of polyphenols, ABTS, and DPPH. The difference was up to 38% in the case of comparing T3 with the control turkey sausage without FDAP. This was due to the high TPC and antioxidant activity of FDAP incorporated into turkey sausages. Grispoldi et al. [21] studied TPC and antioxidant activity of Italian salami with apple pomace and found that apple-pomace-fortified salami samples had significantly higher TPC values, ABTS, and radical scavenging activity, evaluated by DPPH assays, than the control.

### 3.6. Color

The color parameters (L, a, and b) of turkey sausages with various levels of FDAP are presented in Table 4. The incorporation of FDAP significantly (*p* ≤ 0.05) affected the color characteristics of the turkey sausages. All color values decreased significantly with the addition of FDAP to turkey sausages. First, the lightness values of turkey sausages varied from 84.29 to 76.26 (control), from 78.18 to 71.35 (T1), from 79.69 to 70.86 (T2), and from 76.50 to 69.38 (T3) during storage. As expected, lightness values were significantly (*p* ≤ 0.05) lower in the treatments with the FDAP group than in the control group. The decreased lightness might be attributed to the color of apple pomace. This decrease in lightness with increasing levels of FDAP was consistent with the findings of Rather et al. [40], who observed higher lightness in control samples than in goshtaba with apple pomace powder. In contrast, the redness values of turkey sausages increased significantly (*p* ≤ 0.05). The three formulations containing FDAP showed less redness than the control, immediately after production and during storage. The redness values of turkey sausages ranged from 5.87 to 6.65 (control), from 4.73 to 5.24 (T1), from 4.43 to 4.81 (T2), and from 4.31 to 4.79 (T3) during storage. The increase in a values during storage could have been due to oxidative changes in meat pigments, which might have been influenced by the antioxidant properties of FDAP. FDAP incorporation and storage time both contribute to the darkening of turkey sausages. However, this trend was in line with the findings of Keska et al. [22], who also reported increased redness in baked meat products treated with FDAP. The yellowness (b) values were significantly (*p* ≤ 0.05) lower in the FDAP-treated sausages than in those of the control. The b value decreased significantly (*p* ≤ 0.05) with increasing levels of FDAP, indicating a reduction in yellow color intensity. During storage, the trend of decreased yellowness in the turkey sausages continued. These results were similar to those reported by Grispoldi et al. [21] for Italian salami.

### 3.7. Texture Profile Analysis

The texture attributes of turkey sausages with various levels of FDAP are presented in Table 5. The reduction in turkey meat and increase in FDAP levels in the formulations resulted in significantly lower (*p* ≤ 0.05) values for hardness, springiness, cohesiveness, gumminess, and chewiness. Specifically, T3 exhibited significantly lower (*p* ≤ 0.05) hardness values (59.7) than T1, T2, and the control (66.17, 64.37, and 76.73, respectively) immediately after production. This reduction in hardness may be attributed to the incorporation of apple pomace, which likely interfered with protein binding, thus decreasing the overall hardness. Similar trends were observed by Huda et al. [18] for mutton nuggets and Verma et al. [17] for low-fat chicken nuggets. Similar to hardness, the springiness values of the sausages decreased significantly (*p* ≤ 0.05) with the addition of FDAP. The springiness values were 0.78 (control), 0.61 (T1), 0.57 (T2), and 0.48 (T3) immediately after production. Han and Bertram [41] suggested that this reduction in springiness could be attributed to the formation of protein–water or protein–protein gelation networks due to the added fibers. Yadav et al. [42] also reported significantly lower springiness scores in chicken sausages incorporating dried apple and carrot pomaces than in the control. In contrast, the cohesiveness values for the control and treated turkey sausages (T1–T3) were similar (*p* ≥ 0.05), with values of 0.29, 0.24, 0.23, and 0.20, respectively, indicating no significant differences across treatments immediately after production. This observation aligned with that of Thangavelu et al. [37], who found no significant difference in cohesiveness values in Irish breakfast sausages incorporated with apple pomace and coffee silverskin powders. Additionally, the gumminess values of the treated turkey sausages were significantly lower (*p* ≤ 0.05) than those of the controls. The gumminess values were 22.53 (control), 15.79 (T1), 14.75 (T2), and 12.07 (T3) immediately after production. Verma et al. [17] also found a decrease in gumminess in low-fat chicken nuggets with increasing levels of apple pomace. Finally, the chewiness values of the control and treated turkey sausages (T1–T3) differed significantly (*p* ≤ 0.05), with values of 17.61 (control), 9.68 (T1), 8.45 (T2), and 5.78 (T3) immediately after production. This finding was consistent with that of Verma et al. [17], who reported significantly higher (*p* ≤ 0.01) chewiness values in control low-fat chicken nuggets than in treated ones. After 7 days of cold storage, the control and treated turkey sausages (T1–T3) became harder until 84.93, 63.23, 68.87, and 63.43, respectively, during storage. There was an increase in springiness among the control and all treated turkey sausages (T1–T3) during chill storage. Among the treatments incorporating apple pomace, T1 and T2 showed no significant differences in gumminess and chewiness values.

### 3.8. Microbiological Analysis

The results of microbiological analyses of turkey sausages with various levels of FDAP are reported in Table 6. Significant differences were observed in the Standard Plate Count (SPC) among the different treatments on days 0 and 7 of storage. On day 0, the control sausages exhibited the highest SPC (2.6 × 10^3^ cfu/g), while the sausages with the highest FDAP content (T3) showed the lowest SPC (9.0 × 10^2^ cfu/g). This immediate reduction in microbial load with increasing FDAP levels suggested that FDAP possesses strong initial antimicrobial properties, likely because of its phenolic compounds and other bioactive constituents known for their antimicrobial activity. By day 7, all treatments showed a significant increase in SPC, which is typical during refrigerated storage, as microorganisms multiply over time. However, the rates of increase and the final microbial counts were markedly different across the treatments. The control sausages again had the highest microbial load (2.06 × 10^4^ cfu/g) by day 7, while the sausages with the highest FDAP content (T3) had the lowest microbial load (0.75 × 10^4^ cfu/g). The treated sausages (T1, T2, and T3) consistently demonstrated lower SPC than the control, indicating that FDAP continued to exert antimicrobial effects during storage.

### 3.9. Sensory Evaluation

The addition of (FDAP) significantly influenced the sensory properties of turkey sausages. The sensory scores for color, flavor, texture, juiciness, taste, and overall acceptability (OAA) on days 0 and 7 are presented in Figure 3. On day 0, the T1 sausages had the highest color score of 7.5 compared to the control and treatments T2 and T3. By day 7, the color scores had continued to decline among all treatments, with T3 having the lowest score of 5.12. The reduction in color scores can be attributed to the brown–yellow pigments in FDAP, which negatively affect their appearance. This aligned with the findings of Yadav et al. [16] and Huda et al. [18], who reported similar trends in chicken and mutton nuggets with dried apple pomace. The flavor scores varied significantly among treatments by day 7; T2 and T3 had the highest flavor scores of 7.5, which were significantly higher than that of the control (5.12). This suggests that FDAP masked meat and fat flavors, which were more pronounced at higher FDAP levels. This masking effect was also observed by Yasarlar et al. [43] in meatballs with incorporated bran. Texture scores decreased with increasing FDAP levels. On day 0, T1 and T2 had texture scores comparable to those of the control, while T3 had significantly lower scores. By day 7, the trend remained similar, with T3 exhibiting the lowest texture score (5.08). This decrease in texture quality was likely due to the reduced protein binding caused by FDAP, which was consistent with the findings of Comer and Dempster [44] and Huda et al. [18]. Juiciness scores also declined with increasing FDAP levels. On day 0, the control and T1 groups had similar juiciness scores, while T2 and T3 were slightly lower. By day 7, the juiciness of T3 was significantly lower (5.37) than that of the control. This reduction could be due to the lower moisture content of FDAP. Verma et al. [17] found no significant difference in juiciness between the control low-fat chicken nuggets and those incorporating apple pulp, which aligned with our observations of lower FDAP levels. The taste scores decreased with increasing FDAP levels. On day 0, T1 and the control had similar taste scores, whereas T2 and T3 had lower scores. By day 7, the taste score of T3 was significantly lower (5.06) than that of the control. This decline may have been due to the additional taste imparted by FDAP, similar to the findings of Younis and Ahmad [20] in buffalo meat patties. OAA scores reflect the combined effects of individual sensory attributes. On day 0, T1 had the highest OAA score (7.5), which was comparable to that of the control. By day 7, T3 had the lowest OAA score (5.37). The decrease in overall acceptability with higher FDAP levels was consistent with the findings of Yadav et al. [34] and Huda et al. [18], indicating that higher FDAP levels negatively affected sensory quality.

## 4. Conclusions

The study successfully developed turkey sausages by incorporating freeze-dried apple pomace (FDAP) at levels of 3%, 5%, and 8%, finding that 3% FDAP produced sausages with physicochemical and sensory properties comparable to those of the control, while enhancing antioxidant activity and fiber content. However, higher FDAP levels led to declines in sensory attributes like texture and overall acceptability. While FDAP is a low-cost source of phenolics and fiber, future research should address its impact on texture and sensory quality at higher inclusion levels and explore its effects on shelf life and overall nutritional value.

## Figures and Tables

**Figure 1 foods-13-02807-f001:**
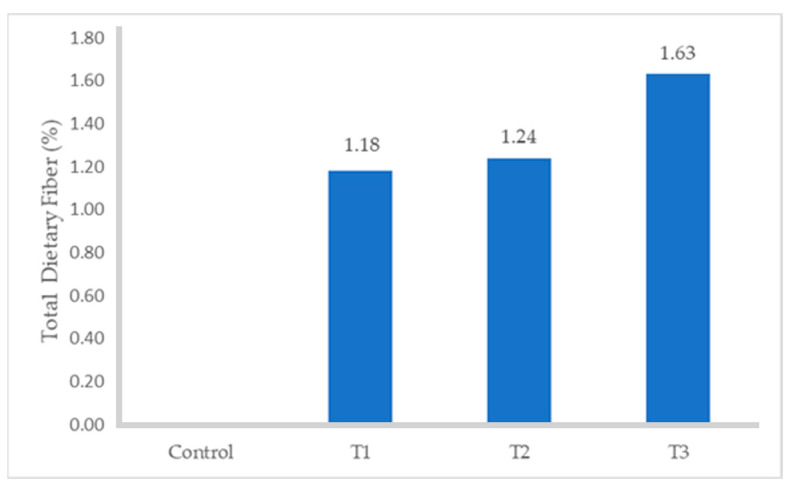
Total Dietary Fiber content of turkey sausage.

**Figure 2 foods-13-02807-f002:**
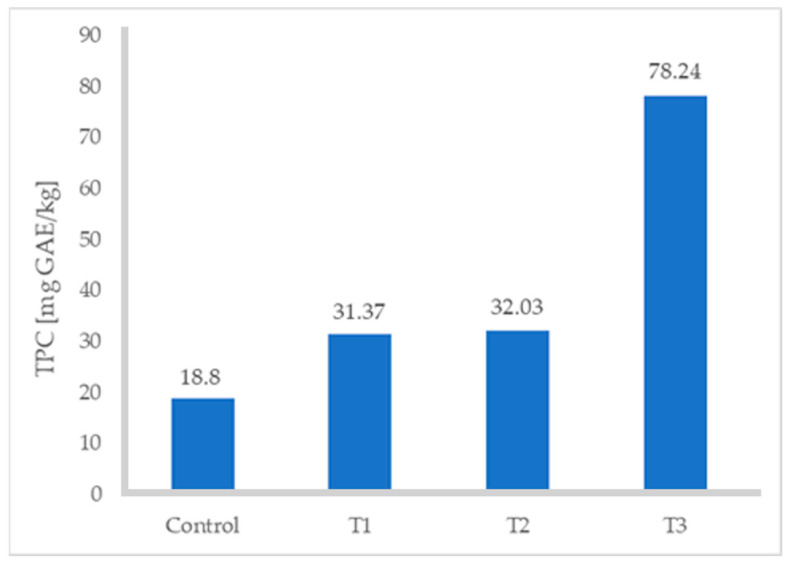
Total phenolic content of turkey sausages.

**Figure 3 foods-13-02807-f003:**
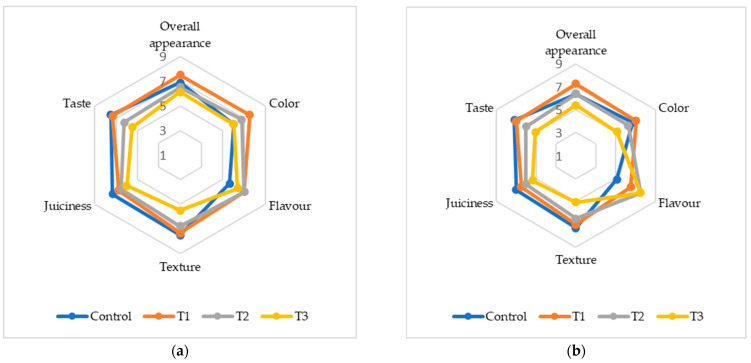
Sensory analysis of cooked turkey sausage formulations with various turkey meat and FDAP levels: (**a**) on day 0; (**b**) on day 7.

**Table 1 foods-13-02807-t001:** Composition and physical properties of freeze-dried apple pomace powder.

Parameter	Value
Moisture Content (%)	8.91
Fat Content (%)	2.45
Protein Content (%)	2.06
Ash Content (%)	2.16
Dietary Fiber Content (%)	21.00
L	76.65
a	7.22
b	17.81
pH	5.29

**Table 2 foods-13-02807-t002:** Turkey sausage formulations with various fat and FDAP levels (units: g/100 g).

Ingredients	Treatments			
	Control	T1	T2	T3
Turkey breast meat	50	47	45	42
Turkey skin	30	30	30	30
Ice water	20	20	20	20
Apple pomace powder	-	3	5	8
Total	100	100	100	100
Curing salt	1.6	1.6	1.6	1.6
Isolated soy protein	1.5	1.5	1.5	1.5
Sugar	0.5	-	-	-

**Table 3 foods-13-02807-t003:** Proximate composition, WHC, cooking loss, and the antioxidant potential of turkey sausage formulations with various turkey meat and FDAP levels.

Treatments	Moisture Content (%)	Protein Content (%)	Fat Content (%)	Ash Content (%)	WHC (%)	Cooking Yield (%)	ABTS (mg TE/g)	DPPH (mg TE/g)
Control	70.93 ± 1.14 ^a^	16.87 ± 0.87 ^a^	9.98 ± 0.68	2.32 ± 0.09 ^a^	75.93 ± 0.75	97.11 ± 0.55	2.53 ± 0.14 ^d^	4.00 ± 0.43 ^c^
T1	70.32 ± 0.79 ^a^	15.35 ± 0.64 ^b^	10.11 ± 0.40	1.87 ± 0.07 ^b^	74.03 ± 2.1	96.71 ± 1.02	2.97 ± 0.18 ^c^	6.09 ± 0.11 ^b^
T2	70.87 ± 0.15 ^a^	14.88 ± 0.75 ^b,c^	10.36 ± 0.61	2.02 ± 0.12 ^b^	75.13 ± 2.53	96.07 ± 0.50	3.61 ± 0.18 ^b^	6.37 ± 0.13 ^b^
T3	68.67 ± 0.16 ^b^	13.89 ± 0.66 ^c^	10.51 ± 0.12	1.95 ± 0.06 ^b^	76.40 ± 1.49	96.48 ± 1.21	4.06 ± 0.33 ^a^	7.02 ± 0.02 ^a^

All values are presented as the mean ± standard deviation of three replicates (n = 3). ^a–d^ Means within a column with different letters are significantly different (*p* ≤ 0.05).; ABTS, 2,2′-azino-bis(3-ethylbenzothiazoline-6-sulfonicacid) diammonium salt; DPPH, 2,2-diphenyl-1-picrylhydrazyl; TE, Trolox equivalents. Control, turkey sausage with 50% turkey breast meat, 20% ice, and without FDAP; T1, turkey sausage with 47% turkey breast meat, 20% ice, and with 3% FDAP; T2, turkey sausage with 45% turkey breast meat, 20% ice, and 5% FDAP; T3, turkey sausage with 42% turkey breast meat, 20% ice and 8% FDAP.

**Table 4 foods-13-02807-t004:** pH and color parameters (L, a, and b) of turkey sausage formulations with various turkey meat and FDAP levels during storage (4 °C).

Time of Storage [Days]	Treatments			
	Control	T1	T2	T3
pH				
0	5.80 ± 0.01 ^a^	5.63 ± 0.02 ^b,c^	5.50 ± 0.02 ^d^	5.34 ± 0.02 ^e^
3	5.83 ± 0.02 ^a^	5.68 ± 0.02 ^b^	5.51 ± 0.02 ^d^	5.35 ± 0.01 ^e^
5	5.85 ± 0.03 ^a^	5.67 ± 0.01 ^b^	5.51 ± 0.01 ^d^	5.36 ± 0.01 ^e^
7	5.85 ± 0.06 ^a^	5.59 ± 0.02 ^c^	5.49 ± 0.04 ^d^	5.31 ± 0.01 ^e^
L				
0	84.29 ± 1.26 ^a^	78.18 ± 0.67 ^b,c^	79.69 ± 1.18 ^b^	76.50 ± 0.48 ^c,d^
3	80.11 ± 0.59 ^b^	75.27 ± 0.36 ^d,e^	76.0 ± 0.43d ^d^	73.53 ± 0.47 ^e^
5	76.63 ± 0.37 ^c,d^	71.30 ± 0.84 ^f^	71.02 ± 1.18 ^f^	69.76 ± 0.36 ^f^
7	76.26 ± 0.45 ^c,d^	71.35 ± 0.36 ^f^	70.86 ± 0.30 ^f^	69.38 ± 0.17 ^f^
a				
0	5.87 ± 0.39 ^b,c^	4.73 ± 0.09 ^d,e,f^	4.43 ± 0.21 ^e,f^	4.31 ± 0.01 ^f^
3	5.87 ± 0.03 ^b,c^	4.81 ± 0.22 ^d,e,f^	4.24 ± 0.13 ^f^	4.35 ± 0.10 ^f^
5	6.27 ± 0.07 ^a,b^	5.23 ± 0.26 ^c,d^	5.030.44 ^d,e^	4.80 ± 0.23 ^d,e,f^
7	6.65 ± 0.22 ^a^	5.24 ± 0.16 ^c,d^	4.81 ± 0.09 ^d,e,f^	4.79 ± 0.19 ^d,e,f^
b				
0	20.91 ± 1.79 ^a^	17.40 ± 0.55 ^c,d^	17.48 ± 0.50 ^c,d^	18.24 ± 0.81 ^b,c^
3	20.2 ± 0.80 ^a,b^	16.35 ± 0.55 ^c,d^	16.93 ± 0.72 ^c,d^	17.59 ± 0.20 ^c,d^
5	17.9 ± 0.86 ^c^	15.6 ± 0.69 ^d^	15.36 ± 0.72 ^d^	16.42 ± 0.13 ^c,d^
7	17.43 ± 0.90 ^c,d^	15.43 ± 0.30 ^d^	16.24 ± 0.28 ^c,d^	16.7 ± 0.37 ^c,d^

All values are presented as the mean ± standard deviation of three replicates (n = 3). ^a–f^ Means within a column with different letters are significantly different (*p* ≤ 0.05).

**Table 5 foods-13-02807-t005:** Texture profile analysis of turkey sausage formulations with various turkey meat and FDAP levels during storage (4 °C).

Time of Storage [Days]	Treatments			
	Control	T1	T2	T3
Hardness [N]				
0	76.73 ± 10.64 ^a,b,c^	66.17 ± 4.61 ^b,c^	64.37 ± 1.53 ^b,c^	59.7 ± 6.04 ^c^
3	71.7 ± 6.77 ^a,b,c^	65.5 ± 2.61 ^b,c^	67.5 ± 9.85 ^a,b,c^	62.7 ± 1.85 ^c^
5	80.5 ± 7.30 ^a,b^	66.57 ± 1.86 ^b,c^	67.37 ± 8.88 ^a,b,c^	61.73 ± 4.85 ^c^
7	84.93 ± 4.65 ^a^	63.23 ± 4.70 ^b,c^	68.87 ± 3.35 ^a,b,c^	63.43 ± 0.49 ^b,c^
Springiness [mm]				
0	0.78 ± 0.00 ^a,b,c,d^	0.61 ± 0.03 ^a,b,c,d^	0.57 ± 0.03 ^b,c,d^	0.48 ± 0.03 ^c,d^
3	0.79 ± 0.03 ^a,b,c,d^	0.74 ± 0.16 ^a,b,c,d^	0.84 ± 0.14 ^a,b,c^	0.53 ± 0.02 ^c,d^
5	0.87 ± 0.08 ^a,b^	0.85 ± 0.12 ^a,b^	0.80 ± 0.17 ^a,b^	0.92 ± 0.00 ^a^
7	0.84 ± 0.07 ^a,b,c^	0.80 ± 0.21 ^a,b,c^	0.83 ± 0.16 ^a,b,c^	0.58 ± 0.08 ^b,c,d^
Cohesiveness [-]				
0	0.29 ± 0.01 ^a,b^	0.24 ± 0.01 ^d,e,f^	0.23 ± 0.01 ^e,f^	0.20 ± 0.01 ^f^
3	0.27 ± 0.03 ^a,b,c,d^	0.25 ± 0.02 ^c,d,e^	0.25 ± 0.00 ^c,d,e^	0.24 ± 0.01 ^d,e,f^
5	0.28 ± 0.01 ^a,b,c^	0.26 ± 0.01 ^a,b,c,d,e^	0.26 ± 0.02 ^a,b,c,d,e^	0.25 ± 0.01 ^c,d,e^
7	0.30 ± 0.01 ^a^	0.27 ± 0.02 ^a,b,c,d^	0.25 ± 0.02 ^b,c,d,e^	0.24 ± 0.01 ^c,d,e,f^
Gumminess [Nm]				
0	22.53 ± 3.60 ^a,b,c^	15.79 ± 1.70 ^d,e^	14.75 ± 0.41 ^d,e^	12.07 ± 1.07 ^e^
3	19.22 ± 2.52 ^a,b,c,d^	16.32 ± 0.45 ^d,e^	16.79 ± 2.53 ^c,d,e^	14.97 ± 1.07 ^d,e^
5	23.02 ± 2.91 ^a,b^	16.92 ± 0.39 ^c,d,e^	17.56 ± 3.59 ^b,c,d,e^	15.25 ± 1.61 ^d,e^
7	25.19 ± 1.43 ^a^	17.04 ± 1.38 ^b,c,d,e^	17.64 ± 2.18 ^b,c,d,e^	15.39 ± 0.45 ^d,e^
Chewiness [N]				
0	17.61 ± 2.89 ^a,b^	9.68 ± 1.27 ^c,d^	8.45 ± 0.31 ^c,d^	5.78 ± 0.32 ^d^
3	15.16 ± 1.87 ^a,b,c^	12.18 ± 2.76 ^b,c,d^	14.01 ± 2.65 ^a,b,c^	8.02 ± 0.82 ^c,d^
5	20.26 ± 4.18 ^a^	14.45 ± 2.05 ^a,b,c^	14.22 ± 4.80 ^a,b,c^	14.02 ± 1.49 ^a,b,c^
7	21.06 ± 2.24 ^a^	13.46 ± 2.79 ^a,b,c,d^	14.83 ± 4.32 ^a,b,c^	8.92 ± 0.87 ^c,d^

All values are presented as the mean ± standard deviation of three replicates (n = 3). ^a–f^ Means within a column with different letters are significantly different (*p* ≤ 0.05).

**Table 6 foods-13-02807-t006:** Standard plate count (cfu/g) of turkey sausage formulations with various turkey meat and FDAP levels during storage (4 °C).

Time of Storage [Days]	Treatments			
	Control	T1	T2	T3
0	2.6 × 10^3^ ± 0.02 ^a^	1.7 × 10^3^ ± 0.03 ^b^	1.3 × 10^3^ ± 0.01 ^b,c^	9.0 × 10^2^ ± 0.001 ^c^
7	2.06 × 10^4^ ± 0.04 ^a^	1.15 × 10^4^ ± 0.08 ^b^	0.89 × 10^4^ ± 0.01 ^c^	0.75 × 10^4^ ± 0.08 ^c^

All values are presented as the mean ± standard deviation of three replicates (n = 3). ^a–c^ Means within a column with different letters are significantly different (*p* ≤ 0.05).

## Data Availability

The original contributions presented in the study are included in the article, further inquiries can be directed to the corresponding authors.

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
