# Peer review of "Utilization and Effect of Apple Pomace Powder on Quality Characteristics of Turkey Sausages"

_foods, 2024, doi:10.3390/foods13172807_

Round 1
Reviewer 1 Report
Comments and Suggestions for Authors
1 Line11: Why did you choose these ratios (3, 5, and 8%)? Whether there is literature to support it or not, please add references.
2. Line26: In key words, there is no ‘functional food’ and ‘textural properties’, ‘sensory evaluation’ are Narrow-minded. Please re-summarize.
3. Line23-24: Please merge the sentences ‘Apples, particularly the domestic variety……in the year 2022’and ‘In 2022, Kazakhstan……267 thousand tons’. There are too many such mistakes in the text, and it is highly recommended to have a polishing service, especially some grammatical errors.
4. Line40: Punctuation superfluous before reference.
5. Line55: Delete the sentence ‘The growing interest in sustainable food……of various meat products’, and put the sentences ‘For example, studies have……salami and baked meat products’ in the third paragraph, please.
6. Line71: Apple pomace powder preparation.
7. Line 79-86: Suggest that making tables.
8. Line 107-110: Not necessary, please delete.
9. Line 138: What does "AOAC" mean? Full name should be given on first occurrence.
10. Line 142: L*, a*, b* should be in italics, please check the full text. There should be spaces before ‘and’.
11. Line 151: Be careful with abbreviations at the beginning of paragraphs, please write the full name.
12. Line 185: The unit (grams) should be consistent in full-text. Please supplement the centrifuge speed.
13. Line 231: ‘p’ should be italicized. Please check the full text. Delete the sentence ‘as the control sample had no additives.’
14. Line 238: Result and discussion. Please describe more in detail, avoiding using consistent with the results or similar as…similar by explain the results of mentioned studies.
15. Line 257: Table 2 & 4 should be merged. Table 3 & 5 should be merged.
16. Line 259: Just keep one and delete the others. Such as line 313, 377 et al.
17. Line 268: Why not make a column chart? The same applies hereinafter.
18. Line 306: Keska et al. Delete punctuation.
19. Line 446: The header is cfu/g, while the content is scientific notation. Please be consistent.
20. Line 498: Delete the sentence ‘Freeze-dried apple pomace was……during storage were assessed’. Please conclude! Do not repeat your results! Just answer on your aim. Besides, please write about the shortcomings of this technology for future generations to study.
Comments on the Quality of English Languageit is highly recommended to have a polishing service, especially some grammatical errors
Author Response
We have marked our revisions in Green line through the article.
Comment 1 Line 11: Why did you choose these ratios (3, 5, and 8%)? Whether there is literature to support it or not, please add references.
Response: Thank you for pointing this out. We selected non-linear ratios of 0%, 3%, 5%, and 8% for apple pomace powder addition because previous work on chicken sausages (Yadav et al., 2016) using linear proportions (0%, 3%, 6%, and 9%) demonstrated that higher levels of dried apple pomace negatively impacted texture and color properties, prompting a focused examination at lower, more relevant concentrations.
Comment 2. Line 26: In key words, there is no ‘functional food’ and ‘textural properties’, ‘sensory evaluation’ are Narrow-minded. Please re-summarize.
Response: Agree. We have changed the keywords to physico-chemical properties, texture profile analysis, apple pomace, cooked sausages, functional food in line 26-27.
Comment 3. Line 23-24: Please merge the sentences ‘Apples, particularly the domestic variety……in the year 2022’and ‘In 2022, Kazakhstan……267 thousand tons’. There are too many such mistakes in the text, and it is highly recommended to have a polishing service, especially some grammatical errors.
Response: Agree. We have merged the sentences ‘Apples, particularly the domestic variety……in the year 2022’and ‘In 2022, Kazakhstan……267 thousand tons’ and deleted the sentence ‘Notably, Kazakhstan serves as the original center for apple cultivation, boasting 35.7 thousand hectares of cultivation area [2]’ in line 31-32.
Comment 4. Line 40: Punctuation superfluous before reference.
Response: Agree. We have accordingly, deleted the punctuation before the reference [3] in Line 36.
Comment 5. Line 55: Delete the sentence ‘The growing interest in sustainable food……of various meat products’, and put the sentences ‘For example, studies have……salami and baked meat products’ in the third paragraph, please.
Response: Agree. We have accordingly, deleted the sentence ‘The growing interest in sustainable food……of various meat products’, and put the sentences ‘For example, studies have……salami and baked meat products’ in the third paragraph in line 52.
Comment 6. Line71: Apple pomace powder preparation.
Response: Agree. We have changed ‘Plant material preparation’ to ‘Apple pomace powder preparation’ in line 66.
Comment 7. Line 79-86: Suggest that making tables.
Response: Agree. We deleted the information about apple pomace in the text, made a link to Table 1 in line 74 and put Table 1 in line 77. Therefore, the orders and names of tables have changed during the article. Number of the Table 2 in line 90, 100, Table 3 in line 221, 248, 268, 311, Table 4 in line 292, 306, 328, Table 5 in line 354, 388, Table 6 in line 393, 408.
Comment 8. Line 107-110: Not necessary, please delete.
Response: Agree. We have accordingly, deleted the note ‘*Control, turkey sausage with 50% turkey breast meat, 20% ice, and without FDAP; T1, turkey sausage with 47 % turkey breast meat,20% ice, and with 3% FDAP; T2, turkey sausage with 45% turkey breast meat, 20% ice, and 5% FDAP; T3, turkey sausage with 42% turkey breast meat, 20% ice, and 8% FDAP.’ in line 101.
Comment 9. Line 138: What does "AOAC" mean? Full name should be given on first occurrence.
Response: Agree. We have added the full name of the AOAC in line 129-131.
Comment 10. Line 142: L*, a*, b* should be in italics, please check the full text. There should be spaces before ‘and’.
Response: Agree. We have changed the L*, a*, b* to italics in line 134, 136, 306, 327, 341, 346, 347 and deleted the space before ‘and’ in line 134.
Comment 11. Line 151: Be careful with abbreviations at the beginning of paragraphs, please write the full name.
Response: Agree. We have changed the ‘TPA’ abbreviation to ‘Texture Profile Analysis (TPA)’ in line 142.
Comment 12. Line 185: The unit (grams) should be consistent in full-text. Please supplement the centrifuge speed.
Response: Agree. We put the centrifuge speed ‘at 20000g during 10 min’ and deleted ‘for 10 minutes (20.878 grams)’ in line 175, also we have changed the word ’60-gr’ to ’60-g’ in line 93.
Comment 13. Line 231: ‘p’ should be italicized. Please check the full text. Delete the sentence ‘as the control sample had no additives.’
Response: Agree. We have changed the p value to italics in lines 15, 222, 224, 235, 238, 251, 268, 270, 282, 292, 311, 328, 337, 346, 355, 356, 362, 369, 374, 378, 380, and deleted the sentence ‘as the control sample had no additives.’ in line 222.
Comment 14. Line 238: Result and discussion. Please describe more in detail, avoiding using consistent with the results or similar as…similar by explain the results of mentioned studies.
Response: Agree. We added more sentences ‘The observed decrease in moisture content in the treated turkey sausages, particularly in treatment T3 (68.67%), can be attributed to the inclusion of freeze-dried apple pomace (FDAP) at increasing levels. FDAP has a much lower moisture content (8.91%) compared to turkey meat, which results in a reduction of overall moisture when it partially replaces the meat in the sausages.’ in line 225-230.
Comment 15. Line 257: Table 2 & 4 should be merged. Table 3 & 5 should be merged.
Response: Agree. We have merged Tables 2 & 4 to Table 3 in line 248-249 and Tables 3 & 5 to Table 4 in line 306-307 and changed the names of Table 3 in line 248-249 and Table 4 in line 306-307. Also, we added the abbreviations under the Table 2 ‘TPC, total phenolic content; GAE, gallic acid equivalents; ABTS, 2,2′-azino-bis(3-ethylbenzothiazoline-6-sulfonicacid) diammonium salt; DPPH, 2,2-diphenyl-1-picrylhydrazyl; TE, Trolox equivalents.’ in line 252-254.
Comment 16. Line 259: Just keep one and delete the others. Such as line 313, 377 et al.
Response: Agree. We have kept one note under Table 3 in line 250-258 and deleted other similar notes under Table 4,5 6 and Figure 3.
Comment 17. Line 268: Why not make a column chart? The same applies hereinafter.
Response: Agree. We have changed Figures 1 and 2 to column charts in lines 261-263 and 322-324.
Comment 18. Line 306: Keska et al. Delete punctuation.
Response: Agree. We deleted the punctuation in the sentence ‘Keska et al. also reported a similar trend in baked meat products the incorporation of freeze-dried apple pomace in baked meat products resulted in a gradual decrease in product pH as the level of FDAP increased [23].’ in line 301.
Comment 19. Line 446: The header is cfu/g, while the content is scientific notation. Please be consistent.
Response: Agree. We have converted the data in Table 6 ‘0.26*104, 0.17*104, 0.13*104, 0.09*104’ to data ‘2.6*103, 1.7*103, 1.3*103, 9.0*102’, also in line 395, 403 and 404.
Comment 20. Line 498: Delete the sentence ‘Freeze-dried apple pomace was……during storage were assessed’. Please conclude! Do not repeat your results! Just answer on your aim. Besides, please write about the shortcomings of this technology for future generations to study.
Response: Agree. We deleted the sentence ‘Freeze-dried apple pomace was used as a substitute for turkey breast meat in turkey sausage formulations, and its effects on the physicochemical properties, oxidative stability, color, and microbiological level during storage were assessed’ in line 451 and we made a conclusion ‘The study successfully developed turkey sausages by incorporating freeze-dried apple pomace (FDAP) at levels of 3%, 5%, and 8%, finding that 3% FDAP produced sausages with physicochemical and sensory properties comparable to the control, while enhancing antioxidant activity and fiber content. However, higher FDAP levels led to declines in sensory attributes like texture and overall acceptability. While FDAP is a low-cost source of phenolics and fiber, future research should address its impact on texture and sensory quality at higher inclusion levels and explore its effects on shelf life and overall nutritional value.’ in line 451-458.

Reviewer 2 Report
Comments and Suggestions for Authors
Reviewer's report
The topic discussed in the article entitled Utilization and effect of apple pomace powder on quality characteristics of turkey sausages is very interesting and widely applicable in poultry farming.
During the review, I encountered several minor errors:
- Capitalize the title.
- Introduction, line 30: Latin names of plants should be written in italics. This applies to the entire article
- Introduction, line 32: The sentence has no space before the period. Careful proofreading of the article is required.
- Introduction, lines 34 and 35: Frequent repetition of "substantial" to find the necessary synonym.
- Introduction, line 40: Check where the reference 4 belongs. Careful proofreading of the article is required.
- Introduction, line 67: It would be appropriate to explain why the linear proportion (0, 3, 6, 9) FDAP was not used and why not linear storage days.
- Material and methods, lines 80, 144 and Table 2: In the context of number notation, use number notation with two decimal places (21.00). When specifying L*, a*, b*, asterisks are overwritten.
- Material and methods, lines 86 and 87: Unify the use of % without space throughout the text.
- Material and methods, lines 92, 93 and 94: The sentence needs to be rewritten.
- Material and methods, lines 108, 262, 316,..: There is a space after the comma. Needs correction throughout the text.
- Material and methods, line 129: Edit the formula notation with appropriate spaces (start, equations, multiplication) and end with a period.
- Material and methods, lines 134 and 160: Check the correct unit notation °C. °C units, there is a space between the value and the unit. Needs correction throughout the text.
- Material and methods, line 135: Fix the use of spaces in parentheses.
- Material and methods, line 147: The sentence needs to be rewritten and explain how many measurements were in statistical analyses for one group.
- Material and methods, line 151: The abbreviation (TPA, TE) used for the first time needs an explanation.
- Material and methods, lines 164-177: Unify notation of units (mM, μL) there is a space between the value and the unit. Needs correction throughout the text.
- Material and methods, lines 221-225: In statistical data processing, explain how triplicates were created and how many measurements were used for statistical analyses. It is necessary to define where the p=0.05 value belongs. In the article, separate statistical analyses within a group between individual time intervals and between groups at the same time intervals. Label clearly in the table footer. Use orthogonal polynomial contrasts to test the linear or quadratic effects of FDAP.
- Results and discussion, lines 248, 305, 306, 331,..: The reference number should be next to the name. Needs correction throughout the text
- Results and discussion Tables: Arrange all the tables according to the instructions.
- Results and discussion Figures: Ensure the figures are of better quality and harmonize the numbers (digits) recorded on the scale and in the graph.
- Results and discussion, line 370: Match the significance notation in the text (p < 0.05).
- References, line 639: Checks for meaningful use of special characters.

Author Response
We have marked our revisions in Orange line through the article.
Comment 1 Capitalize the title.
Response: Thank you for pointing this out. Agree. We have capitalized the title in line 2-3.
Comment 2: Introduction, line 30: Latin names of plants should be written in italics. This applies to the entire article
Response: Agree. We have written the name of plant in italics in line 30.
Comment 3: Introduction, line 32: The sentence has no space before the period. Careful proofreading of the article is required.
Response: Agree. After proofreading of the article, we deleted a space before the period in line 31, 4, 128, 228, 274, 314, 384, deleted the ‘*’ in all Tables after the word Treatments in Table 2, 3,4,5 and 6, the sentence ‘A time interval of 30 s was allowed between two compression cycles’ in line 147-148.
We have changed the order of references in line 36-37, words ‘water’ to words ‘moisture’ in line 82, the word ‘smell’ to ‘flavour’ in line 206, ‘flavor’ to ‘falvour’ in line 413-422, the data ‘(68.67±0.16%)’ to ‘(68.67%)’ in line 227, the word ‘values’ to word ‘value’ in line 341, 346, 347, also we have put ‘*’ to second author in line 4 and changed the correspondence author email and telephone number in line 9, and the names of the Tables in Supplementary Materials in line 462-470.
Comment 4: Introduction, lines 34 and 35: Frequent repetition of "substantial" to find the necessary synonym.
Response: Agree. We have changed the word ‘substantial’ to ‘significant’ in line 33.
Comment 5: Introduction, line 40: Check where the reference 4 belongs. Careful proofreading of the article is required.
Response: Agree. We deleted ‘as indicated by numerous studies [4]’ in line 36.
Comment 6: Introduction, line 67: It would be appropriate to explain why the linear proportion (0, 3, 6, 9) FDAP was not used and why not linear storage days.
Response: Agree. We have selected the non-linear ratios of 0%, 3%, 5%, and 8% for apple pomace powder addition because previous work in chicken sausages (Yadav et al., 2016) using linear proportions (0%, 3%, 6%, 9%) demonstrated that higher levels of dried apple pomace negatively impacted texture and color properties, prompting a focused examination at lower, more relevant concentrations and we chose the non-linear storage days of 0, 3, 5, and 7 because these intervals align with critical points in the spoilage dynamics of turkey sausages, capturing rapid quality changes early in storage, which are more relevant for assessing shelf life and consumer safety, rather than extending to less informative later stages.
Comment 7: Material and methods, lines 80, 144 and Table 2: In the context of number notation, use number notation with two decimal places (21.00). When specifying L*, a*, b*, asterisks are overwritten.
Response: We used a number notation with two decimal places in line 81-83, 136, and in Table 3. Agree. We deleted the all asterisks in all text in line 134, 136; 306, 327; 341; 346, 347 and in Table 4.
Comment 8: Material and methods, lines 86 and 87: Unify the use of % without space throughout the text.
Response: Agree. We have deleted the space in line 81-83 and 243.
Comment 9: Material and methods, lines 92, 93 and 94: The sentence needs to be rewritten.
Response: Agree. We changed the sentence ‘Sausages were prepared using ingredients as per the formulation presented in Table 2, and the control and three levels of FDAP treatments were tested. FDAP was incorporated at 3, 5, and 8% levels to replace the turkey breast meat’ to the sentence ‘The following formulations were obtained: C – control sample (0% FDAP); T1 – sausages with 3% addition of FDAP; T2 – sausages with 5% addition of FDAP; T3 – sausages with 8% addition of FDAP (Table 2)’ in line 87-90.
Comment 10: Material and methods, lines 108, 262, 316,..: There is a space after the comma. Needs correction throughout the text.
Response: Agree. We deleted the comma in sentence ‘T3, turkey sausage with 42% turkey breast meat, 20% ice and 8% FDAP’ in line 258.
Comment 11: Material and methods, line 129: Edit the formula notation with appropriate spaces (start, equations, multiplication) and end with a period.
Response: Agree. We created Equation (1) and placed it on line 120.
Comment 12: Material and methods, lines 134 and 160: Check the correct unit notation °C. °C units, there is a space between the value and the unit. Needs correction throughout the text.
Response: Thank you for pointing this out. Agree. We deleted the space in line 71, 86, 93, 96, 97, 103, 116, 125, 150, 194
Comment 13: Material and methods, line 135: Fix the use of spaces in parentheses.
Response: Agree. We deleted a space in line 128.
Comment 14: Material and methods, line 147: The sentence needs to be rewritten and explain how many measurements were in statistical analyses for one group.
Response: Agree. We changed the sentence ‘The values were measured in triplicate and averaged for statistical analysis’ to the sentence ‘Analyses were performed in triplicate in each series (3x3)’ in line 139.
Comment 15: Material and methods, line 151: The abbreviation (TPA, TE) used for the first time needs an explanation.
Response: Thank you for pointing this out. Agree. We have written the full form of TPA in line 142.
Comment 16: Material and methods, lines 164-177: Unify notation of units (mM, μL) there is a space between the value and the unit. Needs correction throughout the text.
Response: Agree. We have put the space between value and the unit in line 156-165.
Comment: Material and methods, lines 221-225: In statistical data processing, explain how triplicates were created and how many measurements were used for statistical analyses. It is necessary to define where the p=0.05 value belongs. In the article, separate statistical analyses within a group between individual time intervals and between groups at the same time intervals. Label clearly in the table footer. Use orthogonal polynomial contrasts to test the linear or quadratic effects of FDAP.
Response: Response: Thank you for pointing this out. Agree. We added more information in Material and Methods part, in line 211-217: ‘All assays were performed in at least three replicates and three measurements (3x3). The results are presented as mean and standard error (mean ± standard error). The differences between the mean values were considered significant at P<0.05 by using Duncan’s multiple range tests. Data were analyzed using one-way (between groups at the same time intervals) and two-way (within a group between individual time intervals) analysis of variance (ANOVA). All analyses were performed using Statistica software ver. 13.0. (StatSoft Inc., Poland).’ Also, we appreciate your suggestion to use "Orthogonal Polynomials" in our paper. However, we ultimately decided to leave the statistical analysis as it is now.
Comment 17: Results and discussion, lines 248, 305, 306, 331,..: The reference number should be next to the name. Needs correction throughout the text
Response: Thank you for pointing this out. Agree. We have added the reference number next to the name of the authors in the line 37, 236, 300, 301, 317 and throughout the text in line 240, 335, 344, 349, 360, 361, 364, 366, 371, 375, 380, 419, 424, 429, 433, 439, 443.
Comment 18: Results and discussion Tables: Arrange all the tables according to the instructions.
Response: We have added all tables in accordance with the instructions.
Comment 19: Results and discussion Figures: Ensure the figures are of better quality and harmonize the numbers (digits) recorded on the scale and in the graph.
Response: Agree. We changed the Radar graphs to graphs in Match style in Figure 3 in line 445-446.
Comment 20: Results and discussion, line 370: Match the significance notation in the text (p < 0.05).
Response: Thank you for pointing this out. Agree. We added the word significantly (p≤0.05) in line 347.
Comment 21: References, line 639: Checks for meaningful use of special characters.
Response: We put the reference in line 583-584 using Zotero and American Society of Agricultural and Biological Engineers style.
